# Generalized Compressed Sensing for Image Reconstruction with Diffusion Probabilistic Models

**Ling-Qi Zhang**                                                    *zhangl5@janelia.hhmi.org*
*Janelia Research Campus, Howard Hughes Medical Institute*

**Zahra Kadkhodaie**                                          *zkadkhodaie@flatironinstitute.org*
*Flatiron Institute, Simons Foundation*

**Eero P. Simoncelli**[*]                                          *eero.simoncelli@nyu.edu*
*New York University*
*Flatiron Institute, Simons Foundation*

**David H. Brainard**[*]                                        *brainard@psych.upenn.edu*
*University of Pennsylvania*

**Reviewed on OpenReview:** *https://openreview.net/forum?id=lmHh4FmPWZ*

## Abstract

We examine the problem of selecting a small set of linear measurements for reconstructing high-dimensional signals. Well-established methods for optimizing such measurements include principal component analysis (PCA), independent component analysis (ICA) and compressed sensing (CS) based on random projections, all of which rely on axis- or subspace-aligned statistical characterization of the signal source. However, many naturally occurring signals, including photographic images, contain richer statistical structure. To exploit such structure, we introduce a general method for obtaining an optimized set of linear measurements for efficient image reconstruction, where the signal statistics are expressed by the prior implicit in a neural network trained to perform denoising (known as a "diffusion model"). We demonstrate that the optimal measurements derived for two natural image datasets differ from those of PCA, ICA, or CS, and result in substantially lower mean squared reconstruction error. Interestingly, the marginal distributions of the measurement values are asymmetrical (skewed), substantially more so than those of previous methods. We also find that optimizing with respect to perceptual loss, as quantified by structural similarity (SSIM), leads to measurements different from those obtained when optimizing for MSE. Our results highlight the importance of incorporating the specific statistical regularities of natural signals when designing effective linear measurements.

## 1 Introduction

The natural world contains inherently high-dimensional spatial and temporal signals, but imaging systems - such as cameras and microscopes - typically capture only partial, lower-dimensional linear measurements of these signals. Maximizing the informativeness of these measurements is critical for many real-world applications (Pruessmann et al., 1999; Lustig et al., 2007; Zhu et al., 2018; Nehme et al., 2020; Deb et al., 2022). This raises a fundamental question: What is the *optimal* set of low-dimensional linear measurements for a given class of signals?

To address this question, we first need to consider the task of reconstructing the original signal from a set of linear measurements, referred to as a "linear inverse problem". In the general case, where the number of

---

[*]These authors contributed equally to this work.
[1]Code available at github.com/lingqiz/optimal-measurement

available measurements is smaller than the dimensionality of the signals to be reconstructed, the problem is underdetermined. Thus, obtaining a solution requires additional constraints that, either explicitly or implicitly, incorporate assumptions about the structure of the signal distribution. In the Bayesian formulation of the problem, the signal structure is expressed as a prior probability distribution over the signal space. This is combined with the measurement constraints (i.e., likelihood function) to obtain a posterior distribution, from which the reconstructed signal is computed, typically by minimizing an expected loss.

In its most common form, one assumes a Gaussian signal prior, and mean squared error (MSE) loss. In this case, the optimal set of linear measurements are the most significant principal components (PCs), and the optimal reconstruction is the linear projection onto those PCs. Although this formulation is foundational, the Gaussian prior describes only the second-order statistical regularities of signals, and falls short of capturing important higher-order dependencies of many natural signal ensembles, particularly photographic images.

More recently, an alternative framework for optimal linear measurements arose from the use of separable priors with sparse (heavy-tailed) marginals. The seminal work by Donoho (2006) in compressed sensing (CS) proved that when the signal lies within a union of subspaces (an idealized form of separable sparse prior), the optimal measurements should be incoherent with the axes of these subspaces, which can be implemented in practice using a set of randomly chosen vectors. In this setting, an iterative non-linear reconstruction can achieve near-optimal recovery (Tropp, 2006). The sparse prior provides a reasonable model for some signal classes, such as medical images, for which compressed sensing leads to significant empirical improvements over PCA methods. This advance highlights the importance of the signal prior in the design of effective linear measurements, as well as for the reconstruction algorithm.

The statistical characterization underlying PCA and CS assumes alignment of signal content with a particular coordinate system, but this assumption does not adequately capture the statistical structure of natural images (Portilla et al., 2003b; Ballé et al., 2016). In support of this, previous work (Weiss et al., 2007) has demonstrated that although random measurements outperform PCs for idealized sparse signals, this does not hold for natural images. Over the past decade, deep neural networks (DNNs) trained for image processing tasks have been able to exploit ever more complex statistical structure, often described in terms of low-dimensional manifolds. Utilizing the priors implicit in these networks has led to substantial performance improvements in solving linear inverse problems (Romano et al., 2017b; Bora et al., 2017; Kadkhodaie & Simoncelli, 2020; Song et al., 2021). These dramatic performance improvements suggest that the DNN-based priors can more faithfully capture the statistical structure of natural images.

Here, armed with the prior implicit in a DNN trained for denoising (hereafter a "denoiser prior"), we re-visit the question of optimal linear measurement for natural images. Specifically, we develop a framework for optimizing a set of linear measurements to minimize the error obtained via non-linear image reconstruction under the denoiser prior. More specifically, our reconstruction algorithm uses a generative diffusion model based on a trained DNN image denoiser. This enables us to apply our method to natural images and to analyze the impact of natural image statistics on the optimal linear measurements. We demonstrate that these measurements (1) vary substantially with the training dataset (e.g., digit vs. face images); (2) vary with the choice of reconstruction loss (e.g., MSE vs. SSIM); (3) are distinct from those of PCA, ICA and CS, with substantially more asymmetrical marginal distributions; and (4) lead to substantial performance improvements over PCA and CS. This work provides yet another example of the impressive improvements that can be achieved by applying modern ML methods to fundamental problems in signal processing. Our findings also establish a critical baseline for evaluating the potential benefits of non-linear measurements and the impact of measurement noise on the reconstruction of natural signals.

## 2 Optimized Linear Measurement (OLM)

### 2.1 Linear inverse problem

Given an image $x \in \mathcal{R}^d$, we express a linear measurement as $m = M_k^T x$, where $M_k \in \mathcal{R}^{d \times k}$, is a measurement matrix, and $m \in \mathcal{R}^k$ is the measurement which provides a partial observation of $x$ (i.e., $M_k$ is low rank, $k < d$), We assume $m$ noise-free. The linear inverse problem is to reconstruct an approximation of the original image from the measurement, $\hat{x}(m)$, where $\hat{x}(\cdot)$ can be nonlinear.

We take a Bayesian statistical approach to solving the inverse problem, in which a prior distribution of the signal, $p(x)$, characterizes the statistical regularities of $x$. Given a partial observation $m$, one can obtain a posterior distribution, $p(x|m)$, and the inverse problem is formulated to minimize an expected loss over this posterior. For squared error loss, the solution is the conditional mean of the posterior, $\hat{x}(m) = \int x p(x|m)\, dx$, and for a "0-1" loss, it is the mode, $\hat{x}(m) = \arg\max_x\ p(x|m)$. These solutions are known as minimum mean squared error (MMSE) and maximum a posteriori (MAP) estimates, respectively. More recently, stochastic sampling approaches for solving inverse problems have emerged, where the reconstruction is not the mean or maximum of the posterior, but a high-probability sample (Kadkhodaie & Simoncelli, 2020; Kawar et al., 2022; Chung et al., 2022). We describe the stochastic solution in more detail next.

## 2.2 Image prior embedded in a denoiser

Traditionally, image priors were constructed by using simple parametric forms (Geman & Geman, 1984; Lyu & Simoncelli, 2008; Zoran & Weiss, 2011). Improvements in these priors led to steady progress over several decades. Over the last decade, however, the emergence of deep learning has made it possible to *learn* sophisticated priors from data. In particular, score-based diffusion models have exhibited incredible success in drawing samples from learned image priors. Score-based diffusion models are deep neural networks trained to remove Gaussian white noise by minimizing the mean squared error between the clean and denoised images. The learned denoiser is applied partially and iteratively, starting from a sample of Gaussian noise to generate an image. The generated image is a sample from the image prior embedded in the denoiser. The connection between denoising function and prior is made explicit in Tweedie's equation (Miyasawa, 1961; Robbins, 1992; Raphan & Simoncelli, 2007; Vincent, 2011; Efron, 2011):

$$\hat{x}(y) = y + \sigma^2 \nabla \log p_\sigma(y) \tag{1}$$

where $y$ is the noise-corrupted signal: $y = x + z$, $z \sim \mathcal{N}(0, \sigma^2 \mathrm{I})$, and $\hat{x}(y)$ is the MMSE denoising solution. This remarkable equation provides an explicit connection between denoising and the density of the noisy image $p_\sigma(y)$. See Appendix A for the proof of Equation (1). The distribution of noisy images, $p_\sigma(y)$ is related to the image prior $p(x)$ through marginalization:

$$p_\sigma(y) = \int p(y|x)\, p(x)\, dx = \int g_\sigma(y - x)\, p(x)\, dx, \tag{2}$$

where $g_\sigma(z)$ is the distribution of Gaussian noise with variance $\sigma^2$. This equation expresses the convolution of $p(x)$ with the Gaussian probability density function. That is, $p_\sigma(y)$ is a blurred version of $p(x)$ where the extent of blur depends on $\sigma$. The family of $p_\sigma(y)$ over all $\sigma > 0$ forms a scale-space representation of $p(x)$ and is akin to the temporal evolution of a diffusion process of $p(x)$. A learned denoiser trained over a wide range of $\sigma$ approximates this family of gradients of log densities and can then be used in a coarse-to-fine gradient ascent algorithm to sample from $p(x)$ (see Algorithm 1).

## 2.3 Inverse problem as constrained sampling

To utilize the prior for solving inverse problems, the diffusion sampling algorithm can be modified to handle linear constraints. To draw samples from the denoiser prior given the partial linear measurements $m = M_k^T x$, the score of the conditional distribution, $\nabla_y \log p_\sigma(y|m)$, is used instead. The *conditional* score can be written as the following partition (Kadkhodaie & Simoncelli, 2021):

$$\nabla_y \log p_\sigma(y|m) = M_k(m - M_k^T y)/\sigma^2 + (I - M_k M_k^T)\nabla_y \log p_\sigma(y). \tag{3}$$

See Algorithm 2 for a detailed description of the constrained sampling algorithm.

Images obtained using this algorithm are high-probability samples that are consistent with the measurements from the prior embedded in the learned denoiser. Notice that these sampling-based solutions to the inverse problem are not unique, and do not typically minimize mean square error: the MMSE estimate is a convex combination of these samples, and thus will not generally lie on the manifold of natural images from which the conditional samples are drawn. The MMSE estimator is

$$h(m; M_k) \approx \mathbb{E}_{x|m}[x] \tag{4}$$

which can be approximated by averaging over multiple conditional samples for a given measurement model $M_k$, see Appendix E.3 for more details. Visually, however, individual samples will look sharper and of higher visual quality compared to the MMSE estimate (Kadkhodaie & Simoncelli, 2021; Kawar et al., 2022).

## 2.4 Optimized linear measurement

We solve numerically for the set of $k$ linear measurements that minimize the average error of reconstruction through conditional sampling of the posterior. We define a loss function to measure the performance of our MMSE estimate, $h(m; M_k)$:

$$\mathcal{L}(M_k) = \mathbb{E}_x \left[ \; ||h(M_k^T x; M_k) - x||^2 \; \right]. \tag{5}$$

which is approximated by averaging over a training set of images. The Optimized Linear Measurement (OLM) matrix, is computed by minimizing the loss:

$$M_k^* = \underset{M_k : M_k^T M_k = I}{\text{argmin}} \; \mathcal{L}(M_k) \tag{6}$$

for a given choice of $k$. Here, without loss of generality, we consider only matrices $M_k$ with orthonormal columns (i.e., $M_k^T M_k = I$).

In order to solve the optimization problem of Equation (6), we use stochastic gradient descent in the space of orthonormal matrices. Specifically, we parameterize the set of matrices $Q \in \mathcal{R}^{d \times k}$ with orthonormal columns using the Householder product, which represents matrices as a sequence of elementary reflections as the following (Trefethen & Bau, 2022; Shepard et al., 2015):

$$Q = H_1 H_2 ... H_k, \text{ where } H_i = I - \tau_i v_i v_i^T \tag{7}$$

Here, each elementary reflector $H_i$ defines a reflection around a plane. Each vector $v_i$ is of the form $[\mathbf{0}, 1, u_i]^T$, with the first $i-1$ elements being zero, and $\tau_i$ is a scale factor: $\tau_i = 2/(1 + ||v_i||^2)$. The collection of $u_i$'s forms a lower triangular matrix of $\mathcal{R}^{d \times k}$. They are the free parameters $\phi$ of the parameterization $Q(\phi)$. We can thus search for measurement matrices in the space of $\phi$.

We re-write our empirical objective function from Equation (5) using the parameterization $Q(\phi)$:

$$\mathcal{L}(\phi) = \frac{1}{N} \sum_{i=1}^{N} ||h(Q(\phi)^T x_i; \; Q(\phi)) - x_i||_2^2. \tag{8}$$

We search for the optimal measurement matrix within $Q(\phi)$ through variants of gradient descent (Adam optimizer (Kingma & Ba, 2014), see Supplementary Appendix E):

$$\phi_{t+1} \leftarrow \phi_t - \lambda \cdot \nabla_{\phi_t} \mathcal{L}(\phi_t) \tag{9}$$

In practice, the gradient $\nabla_{\phi_t} \mathcal{L}(\phi_t)$ is approximated by computing the MSE (Equation (8)) on a subset of images sampled from the training set on each iteration. The gradient descent formulation is general, and applicable to any differentiable objective. As a demonstration of this, we also explore a perceptual loss function, the structural similarity index measure (SSIM) (Wang et al., 2004), using the implementation in Detlefsen et al. (2022). See Appendix E for details of the datasets, network training, and linear measurement optimization.

# 3 Results

## 3.1 Two-dimensional example

As an illustration of our method, consider a 1-D measurement problem in a 2-D signal space, for three different prior probabilities (Figure 1A): a correlated Gaussian distribution, a K-sparse (union of subspace) model, and a tight distribution along a closed 1D manifold. In each case, we train a small, two-layer fully connected denoising network on the data distribution, which is then used to solve the inverse problem via Algorithm 2.

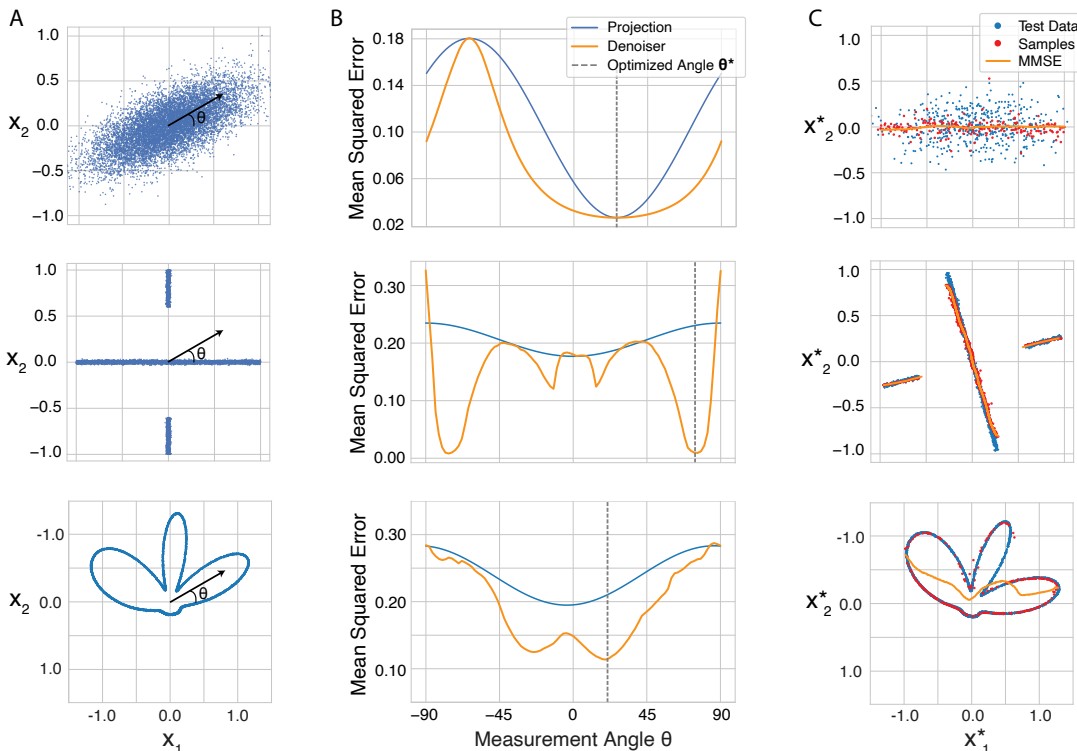

Figure 1: Reconstruction of 2-D signals from a 1-D linear measurement. Top row: Gaussian distribution. Middle row: K-Sparse distribution. Bottom row: A data distribution confined to a curved one-dimensional manifold. **A)** Samples from the target distribution (blue points). Measurements are obtained via dot product between samples and a unit vector (black vector) parameterized by angle $\theta$. **B)** Mean squared error (MSE) of sample reconstruction as a function of measurement angle $\theta$, for two estimators: linear projection onto the measurement vector (blue), and reconstruction using the denoiser prior (orange). The dashed vertical line indicates the optimal solution, $\theta^*$, obtained with our optimization procedure (based on Equation (8)). **C)** Test data drawn from the prior (blue), plotted in a new coordinate system $(x_1^*, x_2^*)$ rotated by $-\theta^*$ (i.e., such that the optimal measurement $m$ corresponds to the horizontal coordinate $x_1^*$). Example bivariate estimates sampled from the denoiser prior conditioned on the optimal linear measurement are plotted in red. The MMSE solution (orange) is obtained by averaging over multiple conditional samples.

The mean squared reconstruction error obtained by this method as a function of the measurement vector angle is shown in Figure 1B (red). As a baseline, we compare this to the reconstruction obtained by linear projection onto the measurement axis (Figure 1B, blue). The optimal measurement vector is obtained by evaluating the reconstruction error for a set of densely sampled 2-D unit measurement vectors spanning orientations $\theta \in [-\pi/2, \pi/2]$ (Figure 1B, orange). For the optimal measurement vector, Figure 1C shows samples conditioned on each measurement value (red points) and the MMSE solution as a function of the measurement value (orange line).

We first consider a bivariate Gaussian distribution (Figure 1, top row). In this case, the first PC is the optimal measurement for both reconstruction methods, as expected for a Gaussian prior. The second row of Figure 1 depicts the result for a union-of-subspace sparse distribution. Here, the first PC is aligned with the horizontal axis. However, the optimal measurement vector for the denoiser prior is dramatically different. The reconstruction error at the optimal $\theta^*$ is near zero. This recapitulates the classical compressed sensing result (Donoho, 2006; Tropp, 2006) that for sparse signals, improved estimates can be achieved by making off-axis measurements.

The last row illustrates the scenario of primary interest in this paper: The data distribution lies in a low-dimensional but curved manifold. This type of higher-order structure cannot be adequately captured by either the Gaussian or sparse prior, but can be effectively described using the more powerful diffusion models, such as our denoiser prior (Supplemental Figure 2). Notably, in this case, we also identified an optimal measurement angle that outperforms the principal axis. Importantly, the optimal angle $\theta^*$ is correctly identified by our optimization method across all three cases, illustrating the generality of our approach.

Finally, we highlight another aspect of the reconstruction demonstrated by the 2D examples (Figure 1C). Our methods generate estimates by sampling from the denoiser prior, conditioned on the linear measurements. While individual samples are consistent with the conditional prior, the MMSE solutions are obtained by averaging multiple conditional samples. This averaging, however, can cause deviations of the MMSE solution from the prior. In the context of image reconstruction, the MMSE solution often exhibits lower visual quality (e.g., blurrier images) compared to individual conditional samples, as it does not lie close to the image prior manifold (Kadkhodaie & Simoncelli, 2021; Kawar et al., 2022). This observation is particularly relevant in the results presented below and further motivates our approach of optimizing measurements with respect to a perceptual loss function later.

## 3.2 Optimized measurements for MNIST

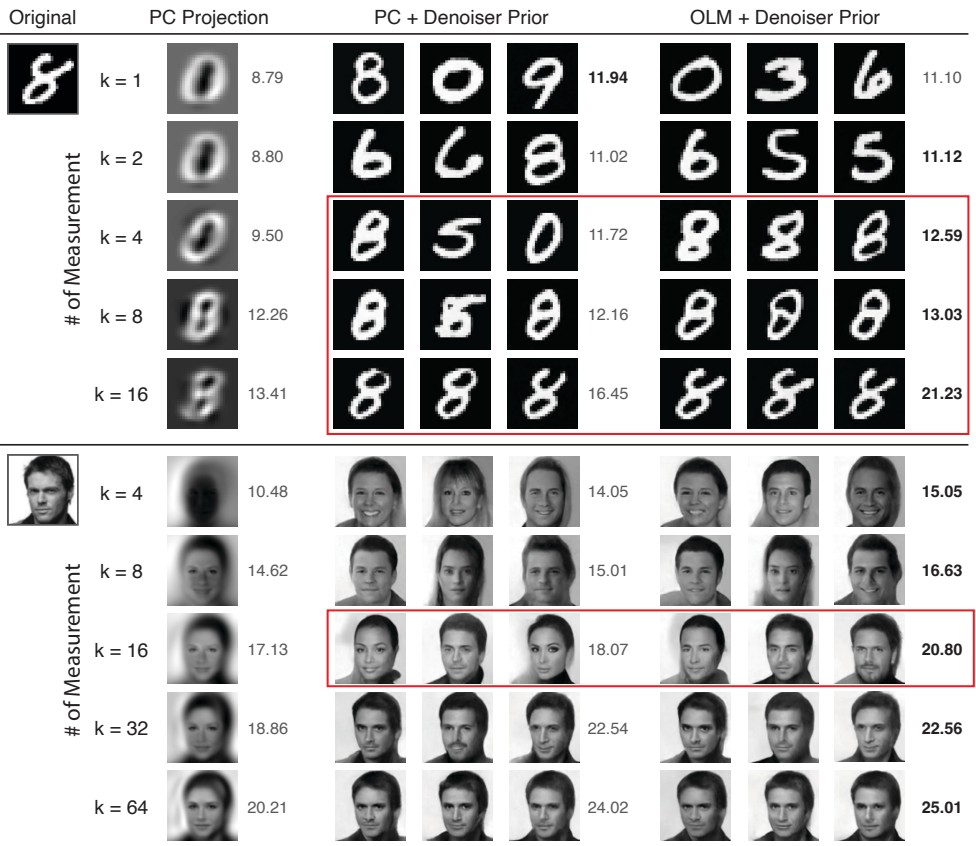

Figure 2: Example image reconstructions using different measurement matrices. Columns show the PC projection, three conditional samples from the denoiser prior using the PCs, and three samples using the OLMs, respectively. Rows correspond to increasing number of measurements, $k$. The numbers indicate the PSNR value of the reconstructions, obtained by averaging over 16 samples. Red boxes highlight examples showing significant visual improvement of the OLM relative to the PC measurements. Top: denoiser trained on (and test example drawn from) the MNIST dataset. Bottom: denoiser trained on the CelebA dataset.

We used our method to obtain optimized measurements for the MNIST dataset (Deng, 2012). We first trained a DNN denoiser on these digit images – for details of the architecture and training, see Appendix E. We apply our method as described in Section 2.4 to obtain the optimized measurement matrix (OLM) for this dataset, for a range of $k$ values.

For comparison, we apply our reconstruction method to two other types of linear measurements: the top $k$ PCs, and $k$ random vectors, which are optimal measurements under Gaussian and sparse prior assumptions, respectively. We also compare to the optimal reconstruction for a Gaussian prior, which is simply linear projection onto the space spanned by the top $k$ PCs. The top half of Figure 2 illustrates our results for a test digit image. First, we observe that combining the PCs with the denoiser prior significantly improves the linear inverse estimates, which reflects the power of the denoiser prior. All conditional samples from the denoiser prior appear to be real MNIST images, which then gradually converge to the original image as $k$ increases. Use of the optimal measurement matrix (OLM) offers additional improvements, reconstructing the correct digit with a smaller $k$. These improvements are evident in both the identity of the digits and their more detailed appearance. Using random measurements (not shown) does not perform as well as either the optimized linear measurements or PCA. See Supplemental Figure 3 for two more examples, including random measurement vectors.

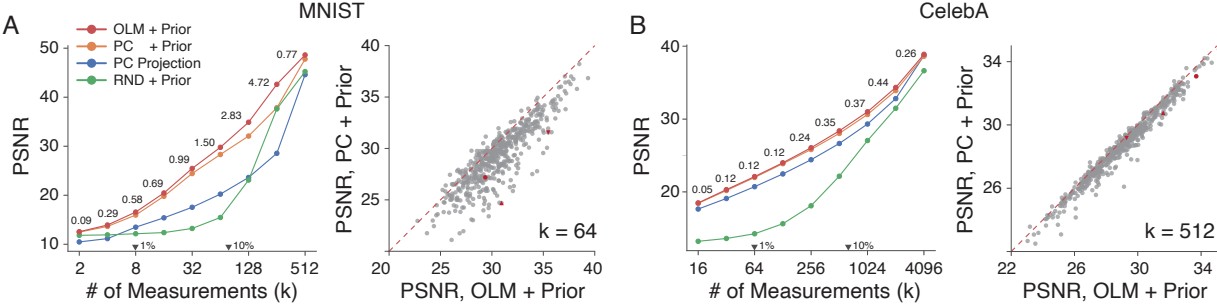

Figure 3: Performance comparisons for **A)** MNIST and **B)** CelebA datasets. Left graph in each panel shows the peak signal-to-noise ratio (PSNR, in dB) as a function of the number of measurements $k$, for the denoiser prior reconstruction from OLMs (red), denoiser prior reconstruction from PCs (orange), linear (projection) reconstruction from PCs (blue), and denoiser prior reconstruction from random measurements (green). Inline numbers indicate the increase in PSNR from PC + denoiser prior to OLM + prior. The right graph in each panel plots the PSNR of the PC reconstruction against that of the OLM reconstruction over all images in the test set, for a value of $k$ approximately 10% of the total number of pixels. Red points indicate the examples shown in Figure 2 and Supplemental Figure 3 and Figure 4. Also see Supplemental Table 1 for comparison of our results to other previous methods.

We quantify our results in terms of peak signal-to-noise ratio (PSNR), as a function of $k$ in the left panel of Figure 3A. The OLMs result in superior reconstruction (red curve), particularly for the values of $k$ ranging from 1% to 30% of the total number of pixels. The peak performance gain of OLM over PC (both utilizing denoiser prior) reached 4.72 dB in PSNR at $k = 256$. Between the two PC reconstructions, the nonlinear prior-based method (orange) shows a considerable improvement over linear reconstruction (blue), demonstrating the substantial advantage of using the more complex denoiser prior. See Supplemental Table 1 for a comparison of our results to other methods in the literature. Lastly, consistent with previous reports using simpler image priors (Weiss et al., 2007), we observe that PC measurements outperform random measurements when used with the denoiser prior (green). This raises the possibility that the union of subspace prior assumed in the compressed sensing literature (Donoho, 2006) does not fully capture the structure of these images, since random measurements are expected to perform well if the images indeed lie on a union of subspaces – a structure that we believe the denoiser prior is expressive enough to capture.

In addition to the average performance, we also compare the performance of these methods on individual images in the right panel of Figure 3A, in a scatter plot of the reconstruction errors from measurements using PCs versus those obtained using OLMs. In both cases, nonlinear reconstruction was performed using

the denoiser prior; thus the plot highlights the differences due to the linear measurements. Importantly, we observe improvement in performance for almost all images in the test set for the OLM.

### 3.3 Optimized measurements for CelebA

To test the generality of our method for different image classes, we repeated our experiments on the CelebA dataset (Liu et al., 2015), which consists of about $200,000$ centered face images. We resized all images to $80 \times 80$ and converted them to grayscale. The rest of the procedure is the same as described above. The bottom half of Figure 2 shows an example face image from the test set. Similar to the MNIST case, we observe that using the denoiser prior to reconstruct from PCs significantly improves the linear inverse estimates. Notably, the conditional samples all appear to be realistic face images even for low $k$; this indicates that the denoiser prior adequately captured the statistical structure of this image dataset. As $k$ increases, the reconstructed face images increasingly resembles the original. Optimizing the measurement matrix offers further improvements in the results, in this case most visible for $k = 16$. See Supplemental Figure 4 for two more examples, including reconstructions obtained with random measurements.

The effect of the optimized measurements is quantified in Figure 3B. As with the MNIST dataset, we observe that OLM leads to superior reconstruction using the denoiser prior, illustrated by the red curve in the left panel. The improvements arising from OLM measurements are consistent across all values of $k$, although they are smaller than for the MNIST case. The peak difference in this case is 0.44 dB in PSNR at $k = 2048$. Thus, the performance gain between PCA and OLM also depends on the specific image dataset.

Consistent with results on MNIST, reconstruction from PCs using the denoiser prior results in higher performance than linear reconstruction, and all of these methods outperform reconstruction from random measurements. We confirmed that the improvements from the OLM are nearly universal for all images in our test set (Figure 3B, right panel). For completeness, we also used independent component analysis (ICA, Hyvärinen & Oja 2000) to generate linear measurement vectors. On both datasets, we found that reconstructions using the denoiser prior based on ICA measurements were marginally worse than those using the PCs (not shown).

### 3.4 Characterizing the optimized linear measurements

In this section, we present some qualitative and quantitative analyses to help interpret the difference between the linear measurement subspaces defined by PCs and OLMs. For the MNIST dataset, as expected, the first few PCs appear digit-like, and the measurement vectors contain increasing high-spatial frequency content as $k$ increases (Figure 4A, left). The OLMs, on the other hand, do not follow this coarse-to-fine (low to high frequency) ordering and are visually much more similar to each other (Figure 4A, right). This is reflected, quantitatively, in the measurement variance as a function of $k$, which falls rapidly for the PCs, but slowly for the OLMs (Figure 4C, left).

Similar phenomena are observed for the CelebA dataset (Figure 4B, C). The PCs, known as "eigenfaces" (Sirovich & Kirby, 1987; Turk & Pentland, 1991), contain features that are geometrically aligned with features of the face, with increasing spatial frequency content. The variance of the measurements across PCs falls approximately exponentially. Although the OLMs also have a face-like appearance, they differ from the PCs. Each OLM vector has a detailed representation of fine facial features (eyes, nose, lips), and a coarse representation of the surrounding content (forehead, chin, hair). As in the MNIST case, all OLMs have similar spatial frequency composition. Consistent with this, the variance of the measurements falls slowly across the OLMs (Figure 4C, right).

To quantify the differences between these two sets of measurement vectors, we use the Grassmann distance between the subspaces they span (Hamm & Lee, 2008; Björck & Golub, 1973). Figure 4D shows this distance as a function of $k$. As a control, the blue line measures the distance between subspaces arising from PCs computed on two random halves of the training data. As expected, the distance between these two subspaces is small. On the other hand, for both the MNIST and CelebA dataset, the subspaces spanned by the optimal measurements are distinct from those defined by the PCs (Figure 4D, red), but they are more similar to the PCs than to the space defined by a set of random measurements (Figure 4D, green).

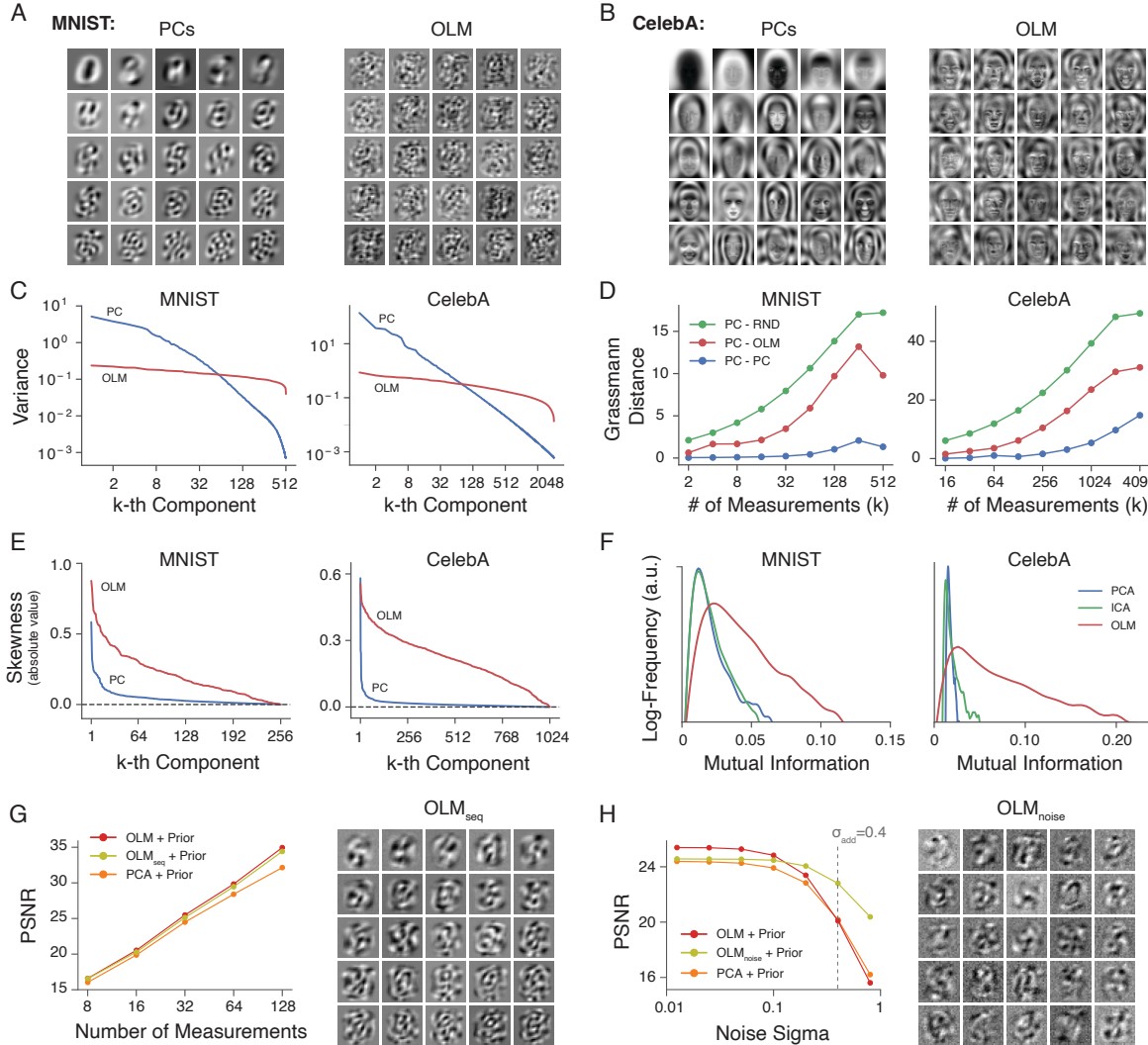

Figure 4: Optimized measurement vectors are distinct from PC and random measurements. **A)** Left: Twenty-five selected principal components of the MNIST dataset, visualized as images, sorted by variance explained from top to bottom, and left to right. Right: Twenty-five components from the columns of the OLM for $k = 64$. **B)** Same as **A)** but for the CelebA dataset. **C)** The variance of the measurement on each component as a function of number of measurements ($k$) for the PCs (blue) and OLMs (red). Note that the optimal measurements are sorted post hoc based on the variance explained by each component. These vectors are optimized jointly for each $k$ and are not ordered by the algorithm itself. **D)** The Grassmann distance between subspaces spanned by different linear measurements as a function of $k$ (Hamm & Lee, 2008). We show the distance between subspaces spanned by PCs obtained on two random halves of the training data (blue), between subspaces spanned by PCs and OLMs (red), and between subspaces spanned by PCs and a set of random measurements (green). See Appendix E for the definition of Grassmann distance. **E)** The (absolute) skewness value of the measurement on each component as a function of number of measurements ($k$, sorted post hoc based on the skewness value) for the PCs (blue) and OLMs (red). **F)** Distribution of mutual information between all pairs of measurement directions for PCA (blue), ICA (green), and OLM (red). **G)** Performance (on MNIST) and visualization of OLMs (olive, OLM$_{seq}$) obtained via a sequential version of the algorithm Appendix E.6. **H)** Performance as a function of measurement noise (on MNIST, $k = 32$) and visualization of OLMs obtained with Gaussian noise ($\sigma = 0.4$) added to the linear measurements during optimization (olive, OLM$_{noise}$), along with comparison to the regular OLMs (red) and PCs (orange).

We can also characterize the linear measurements by analyzing the distribution of measurement values they produce in response to natural images (i.e., the measurement distribution). We found that OLMs differ from traditional methods in two key aspects: First, their measurement distributions are highly asymmetrical, as indicated by the significant non-zero skewness values (see Figure 4E). Second, unlike PCA or ICA, OLMs produce non-factorized measurement distributions: the measurements across axes are not fully independent. This dependency is reflected in the increased pairwise mutual information between different measurement directions observed for OLMs compared to those obtained with PCA or ICA (Figure 4F). This observation aligns with the intuition from the 2D example in Figure 1, where optimal measurements for non-Gaussian priors are not necessarily aligned with the axes.

Lastly, we also consider two modifications to the optimization procedure that result in different OLMs. First, in Figure 4G we present a set of OLMs that are sequentially organized – similar to PCs – using a greedy version of the algorithm (see Appendix E.6), where the measurement subspace for smaller $k$ is always contained within that of larger $k$. We found that while enforcing this sequential constraint results in a slight decrease in performance, the resulting OLMs exhibit more ordered spatial frequency content. Nonetheless, they remain distinct from the PCs both visually and when comparing subspace distances (not shown).

Second, in Figure 4H we examine the robustness of different measurement matrices to noise. For both PCs and OLMs, PSNR decreases as an increasing amount of Gaussian noise is added to the measurements. The OLMs can be made more robust to noise by injecting measurement noise during the OLM optimization (in this case, with a fixed standard deviation of $\sigma = 0.4$). This results in a more noise-robust set of linear measurements (OLM$_{\text{noise}}$), accompanied by a modest performance loss in low-noise regimes. The noise-optimized OLMs are less smooth than those optimized without noise (Figure 4H). A more thorough investigation of noise robustness is worthwhile, but will require the development of a reconstruction algorithm that operates on noisy measurements.

### 3.5 Optimized linear measurements for perceptual loss

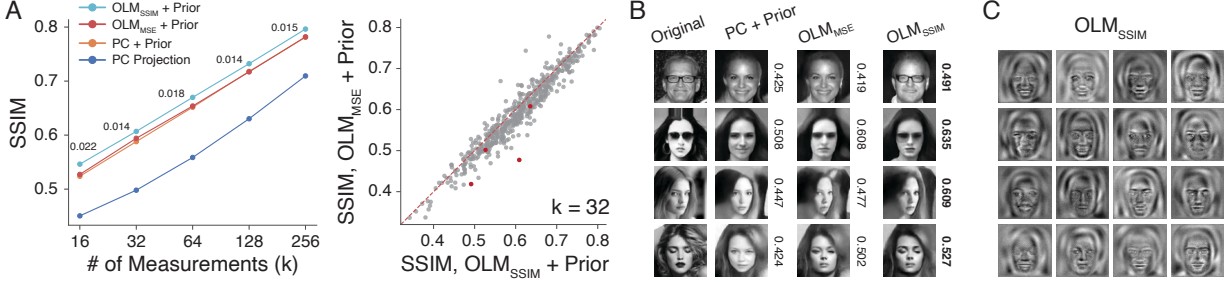

Figure 5: Optimal measurements for perceptual loss, as measured by the Structural Similarity Index Measure (SSIM, Wang et al. (2004)). **A)** Left: SSIM as a function of number of measurement $k$, for PC projection, PC, and OLM optimized for MSE and SSIM, all three using the denoiser prior. The number indicates the increase in SSIM from OLM optimized for MSE to OLM optimized for SSIM. Right: Scatter plot of the SSIM value for individual images, for OLM optimized for SSIM (x-axis) and MSE (y-axis), respectively. **B)** Example linear inverse solutions obtained from different measurement matrices (at $k = 32$) combined with the denoiser prior. The numbers indicate the SSIM value for each individual image. **C)** Example measurement vectors from the OLM optimized for SSIM.

The previous sections described measurements optimized for MSE. However, our method is general, and can be used to optimize any differentiable objective function defined on the linear inverse estimates.

As an example, we computed a set of OLMs with respect to structural similarity index (SSIM) (Wang et al., 2004), a widely used perceptual image quality metric. Figure 5 shows results obtained using this set of optimal linear measurements. We obtained a noticeable improvement in mean SSIM for a range of $k$, and this improvement applies to the majority of the images in the test set. In Figure 5B, four example images are shown to highlight the visual improvement obtained by optimizing for the SSIM objective. However, it is

worth noting that optimizing for SSIM does result in a lower average PSNR compared to the PCs, suggesting a trade-off between optimizing for perceptual quality versus MSE reconstruction, at least with our methods.

Lastly, in Figure 5C, the SSIM-optimized measurement vectors are visualized, which appear quite distinct from both the PCs and their MSE-optimized counterpart shown in Figure 4B. We further confirmed this difference by analyzing the measurement subspace, and found that the subspace distance between OLMs optimized toward different objectives is comparable to their distance from the PCs.

## 4   Related Work

In this section, we discuss recent developments in specifying natural image priors as well as previous approaches to optimizing linear measurements, including analyses of cameras and biological vision.

**Image Priors.** Traditionally, prior models have been developed by combining constraints imposed by structural properties, such as translation or dilation invariance, with simple parametric forms, such as Gaussian, mixtures of Gaussian, or local Markov random fields. These models have been used for solving inverse problems with steady and gradual improvement in performance (Donoho, 1995; Simoncelli & Adelson, 1996; Moulin & Liu, 1999; Hyvärinen, 1999; Romberg et al., 2001; Şendur & Selesnick, 2002; Portilla et al., 2003a; Lyu & Simoncelli, 2008; Zoran & Weiss, 2012). More recently, methods such as VAEs (Kingma & Welling, 2013) and GANs (Goodfellow et al., 2020) have been developed to learn more complex image priors by taking advantage of the increased expressivity of deep neural networks. In the last few years, score-based diffusion models have emerged as the state-of-the-art method for learning sophisticated image priors (Sohl-Dickstein et al., 2015; Song & Ermon, 2019; Ho et al., 2020; Nichol & Dhariwal, 2021), as evidenced by the high quality of generated images obtained as draws from these priors. Diffusion models rely on the explicit relationship between the MMSE denoising solution and the score of the noisy image density (Miyasawa, 1961; Raphan & Simoncelli, 2011; Vincent, 2011). In addition to enabling unconditional sampling from the implicit prior, the power of that prior can be utilized to obtain high-quality solutions to inverse problems (Kadkhodaie & Simoncelli, 2020; Kawar et al., 2022; Wang et al., 2022; Chung et al., 2022; Zhang et al., 2024; Li et al., 2024). A closely related line of work known as *plug-and-play* (P&P) (Venkatakrishnan et al., 2013) used denoisers as regularizers for solving inverse problems. A number of recent extensions have used this concept to develop MAP solutions for inverse problems (Chan et al., 2016; Romano et al., 2017a; Zhang et al., 2017; Kamilov et al., 2017; Meinhardt et al., 2017; Chan et al., 2017; Mataev et al., 2019; Teodoro et al., 2019; Sun et al., 2019; Reehorst & Schniter, 2019; Pang et al., 2020; Sun et al., 2023).

**Camera and Sensor Design.** The optimal linear measurement problem has arisen in studying the design of cameras or sensory systems. In this case, the sensor measurements are also typically linear, but constrained to be positive-valued and spatially localized. Hardware design considerations often impose additional constraints, for example, that the sensor array must be periodic in its structure. For example, Levin et al. (2008) evaluated the impact of different choices in camera design through light field projections in a Bayesian framework. Manning & Brainard (2009) enumerated all possible arrangements of a small one-dimensional sensory array to understand the trade-off between spatial and chromatic information, again in the context of a Bayesian framework. Similarly, an important line of work has examined the problem of optimizing the measurements made by retinal circuits using an information-theoretical objective, in some cases combined with biophysical constraints (Atick & Redlich, 1992; Karklin & Simoncelli, 2011; Jun et al., 2021; Roy et al., 2021; Jun et al., 2022; Zhang et al., 2022). The recent development of differentiable rendering has also allowed for end-to-end optimization of optical systems (Tseng et al., 2021; Deb et al., 2022).

**Optimized Linear Measurements.** Other work has considered the problem of optimizing linear measurements directly. Weiss et al. (2007) observed that neither PCA nor random measurement fully takes into account the statistics of natural images. They attempted to optimize linear measurement using information-based criteria, but were not able to find measurements that outperformed PCA. Wu et al. (2019b) also proposed a method to find better measurement functions (i.e., both linear and nonlinear) in a compressed sensing framework by using the restricted isometry property (Candes, 2008) as an objective. They showed that the optimized measurement functions (both linear and nonlinear) are superior to simple random measurements, but did not compare their results with PCA. They also proposed an optimization method for finding additional structure in signals beyond sparsity (Wu et al., 2019a). Pineda et al. (2020) proposed a

method to improve sampling efficiency in an fMRI setting through reinforcement learning. In a different line of work, Burge & Geisler (2015); Herrera-Esposito & Burge (2025) developed a method for finding linear measurements that are optimal for specific downstream tasks, such as estimating the motion presented in a video sequence.

The main advantages of our method are that (1) the linear measurements are optimized directly with respect to the end objective of the reconstruction problem, and that (2) we incorporate a highly expressive learned prior to exploit the higher-order statistics of natural images. As was shown in Section 2, our method outperforms both random measurements and PCA.

## 5  Conclusion

We present a method for finding optimal linear measurements, using a nonlinear reconstruction method based on a learned prior embedded in a denoiser. We show numerically that the set of linear measurements found through our method results in better image reconstruction than PC measurement, despite the fact that the latter captures more signal variance. The key components of our method are: (i) a denoising diffusion model, which allows us to learn a complex prior underlying the datasets; (ii) a linear inverse algorithm that uses the diffusion prior to generate image reconstructions from linear measurements; and (iii) an end-to-end optimization procedure to find the measurement matrix that minimizes a loss function on the reconstructed images. We show that the optimal measurements are sensitive to both the statistics of the image dataset, and to the objective function for which they are optimized. Our results highlight the importance of accurately modeling the statistics of the signals to design efficient linear measurements.

One major limitation of our current work is that optimizing linear measurements through the iterative diffusion process is substantially more computationally expensive than computing PCs or generating random CS measurements. Additionally, the OLMs are obtained separately for each $k$, and are therefore not sequentially ordered like the principal components. We have restricted ourselves to linear measurements to maintain a direct connection to classical signal processing literature, as they are easier to understand and visualize. Despite this simplifying assumption, it is difficult to ascertain the relationship of the learned measurements to the diffusion prior, and thus to the reconstructed images. It is of interest to expand the current results to linear measurements with realistic noise models, or additional measurement constraints such as locality or non-negativity. Finally, developing reconstruction methods that couple DNN priors with more general optimized nonlinear measurements presents a challenging but enticing goal for future research.

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

## Appendix

## A Tweedie / Miyasawa Proof

For completeness, we provide the proof of the core relationship underlying diffusion models (Miyasawa, 1961; Robbins, 1992; Efron, 2011; Raphan & Simoncelli, 2007). Consider the problem of estimating a natural image $x$ given a noise-corrupted measurement $y = x + \epsilon$, with $\epsilon \sim \mathcal{N}(0, \sigma^2 I)$. The estimate $\hat{x}(y)$ is chosen to minimize the mean of the squared error $||x - \hat{x}(y)||_2^2$ over all images $x$ and observations $y$ corrupted with known $\sigma^2$. In Bayesian terms, the solution to this denoising problem is the posterior mean:

$$\hat{x}(y) = \int x \; p(x|y) \; dx = \int x \; \frac{p(y|x)p(x)}{\int p(y|x)p(x)dx} \; dx. \tag{10}$$

Miyasawa (1961) showed that the optimal estimator has a direct relationship to the score function of $\nabla \log p(y)$. First, write $p(y) = \int p(y|x)p(x)dx$. Assuming a Gaussian PDF for $p(y|x)$ and taking the gradient with respect to $y$ gives:

$$\nabla_y p(y) = \frac{1}{\sigma^2} \int (x - y)p(y|x)p(x)dx. \tag{11}$$

Dividing both sides by $p(y)$ yields:

$$\begin{aligned} \sigma^2 \frac{\nabla_y p(y)}{p(y)} &= \int (x - y)\frac{p(x, y)}{p(y)}dx \\ &= \int (x - y)p(x|y)dx \\ &= \int xp(x|y)dx - \int yp(x|y)dx. \end{aligned} \tag{12}$$

Simplifying this gives the result:

$$\sigma^2 \nabla_y \log p(y) = \hat{x}(y) - y. \tag{13}$$

Thus, the residual of the optimal denoiser, $\hat{x}(y) - y$ is proportional to the gradient of $\log p(y)$. Note that $p(y)$ is equal to the signal prior $p(x)$ convolved with (blurred by) a Gaussian with variance $\sigma^2$. This quantity can be used to sample from the implicit prior $p(x)$, for example, by using annealed Langevin dynamics, where the noise $\sigma^2$ corresponds to different temperature levels Song & Ermon (2019); Bussi & Parrinello (2007); Bhattacharya & Waymire (2009). See Appendix D for a detailed description of our sampling procedure.

## B Principal Components Analysis

Denote by $X$ the data matrix where the columns are individual mean-subtracted samples $x_i$. The linear projection is defined as $\hat{x}_i = MM^T x_i$, where $M$ is a matrix with orthogonal columns. Plugging this expression into Equation (5), we have the expected squared-error loss for linear projection:

$$\begin{aligned} l(M) &= \frac{1}{N} \sum_i ||x_i - MM^T x_i||_2^2 \\ &= \frac{1}{N} \; \text{Tr}[(X - MM^T X)^T (X - MM^T X)] \\ &= \frac{1}{N} \; (\text{Tr}[X^T X] - \text{Tr}[M^T XX^T M]) \\ &= \frac{1}{N} \; (\text{Tr}[X^T X] - \sum_i [m_i^T XX^T m_i]) \end{aligned} \tag{14}$$

To minimize $l(M)$, the $m_i$'s should be the eigenvectors associated with the largest eigenvalues of $XX^T$, which is the empirical covariance matrix. Note that this solution does not assume a Gaussian data distribution:

For any data, the optimal (MMSE) linear projection is the projection onto the leading principal components of the covariance matrix.

When the data is in fact Gaussian distributed, linear projection onto the first $k$ PCs does achieve the minimal error possible, as shown in the bivariate example in Fig. 1A. For non-Gaussian distributions, nonlinear estimation can potentially improve performance.

## C   Sampling Algorithm

---

**Algorithm 1** Sampling via ascent of the log-likelihood gradient from a denoiser residual

---

**Require:** denoiser $f$, step size $h$, stochasticity from injected noise $\beta$, initial noise level $\sigma_0$, final noise level $\sigma_\infty$, distribution mean $m$

1: $t = 0$
2: Draw $x_0 \sim \mathcal{N}(m, \sigma_0^2 \mathrm{Id})$
3: **while** $\sigma_t \geq \sigma_\infty$ **do**
4: $\quad t \leftarrow t + 1$
5: $\quad s_t \leftarrow f(x_{t-1}) - x_{t-1}$ $\qquad\qquad\qquad$ ▷ Compute the score from the denoiser residual
6: $\quad \sigma_t^2 \leftarrow \|s_t\|^2/d$ $\qquad\qquad\qquad$ ▷ Compute the current noise level for stopping criterion
7: $\quad \gamma_t^2 = \left((1-\beta h)^2 - (1-h)^2\right)\sigma_t^2$
8: $\quad$ Draw $z_t \sim \mathcal{N}(0, I)$
9: $\quad x_t \leftarrow x_{t-1} + hs_t + \gamma_t z_t$ $\qquad\qquad\qquad$ ▷ Perform a partial denoiser step and add noise
10: **end while**
11: **return** $x_t$

---

## D   Constrained Sampling Algorithm

We adopt a previously developed sampling algorithm for solving linear inverse problems (Kadkhodaie & Simoncelli, 2020). Given a trained least-squares denoiser $\hat{x}(y)$, define the denoiser residual $g(y) = \hat{x}(y) - y$, which is equal to the score of the learned implicit distribution, $\nabla_y \log p(y)$. The orthogonal linear measurement matrix is denoted as $M \in \mathcal{R}^{d \times k}$. The parameters of the algorithm are step size $h$, the magnitude of injected noise $\beta$, and a stopping criterion $\sigma_{end}$. The inputs are the linear measurements $m \in \mathcal{R}^k$.

---

**Algorithm 2** Constrained sampling for solving linear inverse problem

---

**Require:** $m$, $g$, $M$, $h$, $\beta$, $\sigma_{end}$

1: $t \leftarrow 1$ $\qquad\qquad\qquad\qquad\qquad\qquad\qquad\qquad\qquad\qquad\qquad$ ▷ initialization
2: $\mu \leftarrow 0.5(\mathbf{1} - MM^T\mathbf{1}) + Mm$
3: $y_1 \leftarrow \mathcal{N}(\mu, I)$
4: $\sigma_1^2 \leftarrow \|g(y_1)\|_2^2/d$

5: **while** $\sigma_t > \sigma_{end}$ **do**
$\quad l_t \leftarrow (I - MM^T)g(y_t) + M(m - M^Ty_t)$ $\qquad\qquad$ ▷ compute the conditional gradient
$\quad \gamma^2 \leftarrow [(1 - \beta * h)^2 - (1-h)^2] * \sigma_t^2$ $\qquad\qquad$ ▷ scale factor for the added noise
$\quad y_{t+1} \leftarrow y_t + h * l_t + \gamma * \mathcal{N}(0, I)$ $\qquad\qquad$ ▷ move up the gradient and add noise
$\quad \sigma_{t+1}^2 \leftarrow \|l_t\|_2^2/d$ $\qquad\qquad$ ▷ compute an estimated noise level
$\quad t \leftarrow t + 1$
6: **end while**

---

For the current paper, we chose $h = 0.1$, $\beta = 0.1$, and $\sigma_{end} = 0.01$.

The impact of these parameters on reconstruction performance, along with the number of conditional samples used to obtain the MMSE, is illustrated in the figure below.

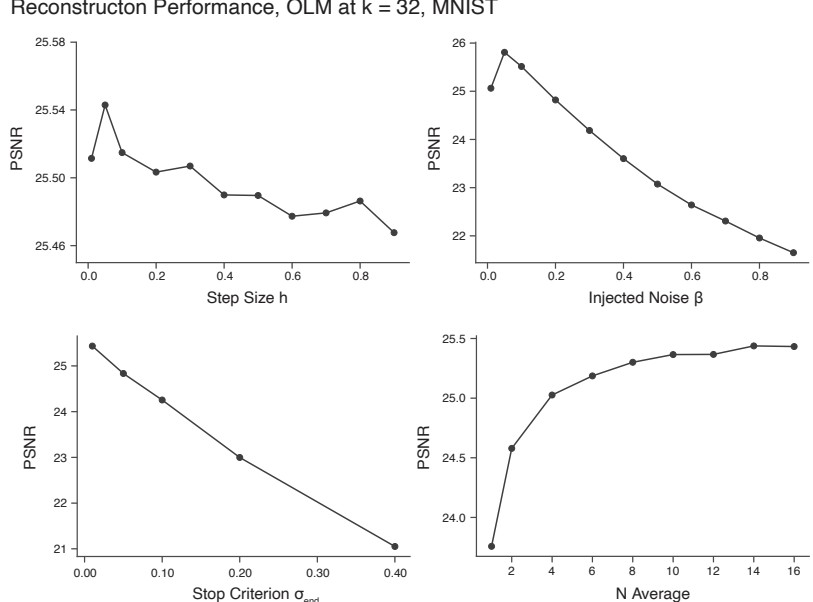

Supplementary Figure 1: The effect of different parameters (step size $h$, injected noise $\beta$, stop criterion $\sigma_{\text{end}}$, and number of conditional samples used for averaging) of the algorithm on reconstruction performance, demonstrated using the OLM at $k = 32$ on the MNIST dataset.

In general, the step size $h$ has a very small effect (note the scale of the y-axis). Achieving good reconstruction performance requires a larger amount of injected noise and a sufficiently small noise stop criterion. Additionally, averaging multiple conditional samples significantly improves performance, but the effect quickly diminishes as $n$ increases.

# E    Experimental Details

## E.1    Dataset

We used two primary two datasets for our experimental results. The CelebA celebrity faces dataset (Liu et al., 2015), which contains approximately $200,000$ images, and the MNIST handwritten digits dataset, which contains $60,000$ images (Deng, 2012). For the CelebA dataset, we downsampled the images to a resolution of $80 \times 80 \times 1$ grayscale in our main results. The images in the MNIST dataset are the original $28 \times 28 \times 1$ grayscale.

In both cases, we randomly selected $n = 512$ images as the test set. The rest of the images were used for training the denoiser, computing the principal components, and optimizing the linear measurement matrix. The test set was only used for reporting the final performance of the optimized linear measurements.

## E.2    CNN denoiser

We performed empirical experiments using UNet architecture. Our UNet networks contain 3 encoder blocks, one mid-level block, and 3 decoder blocks (Ronneberger et al., 2015). (For MNIST images we reduced the encoder and decoder blocks to 2, since they are smaller images.) Each block consists of 2 convolutional layers followed by a ReLU non-linearity and bias-free batch-normalization. Each encoder block is followed by a $2 \times 2$ spacial down-sampling and a 2 fold increase in the number of channels. Each decoder block is followed by a $2 \times 2$ spacial upsampling and a 2 fold reduction of channels. The total number of parameters is $7.6m$. All the denoisers are "bias-free": we remove all additive constants from convolution and batch-normalization

operations (i.e., the batch normalization does not subtract the mean). This facilitates universality (denoisers can operate at all noise levels) – see Mohan et al. (2019).

We follow the training procedure described in Mohan et al. (2019), minimizing the mean squared error in denoising images corrupted by i.i.d. Gaussian noise with standard deviations drawn from the range $[0, 1]$ (relative to image intensity range $[0, 1]$). Training is carried out on batches of size 512, for 1000 epochs. Note that all denoisers are universal and blind: they are trained to handle a range of noise, and the noise level is not provided as input to the denoiser. These properties are exploited by the sampling algorithm, which can operate without manual specification of the step size schedule (Kadkhodaie & Simoncelli, 2020). This method produces high-quality results in generative sampling, as well as sampling conditioned on linear measurements (Kadkhodaie & Simoncelli, 2021). To train each denoiser, 4 NVIDIA A100 GPU were used. The total training time for each denoiser was approximately 10 hours.

### E.3   MMSE estimate

We construct the linear inverse estimate by averaging multiple samples obtained using Algorithm 2. The individual samples are shown in Figure 2 and Supplementary Figure 3 and Figure 4. Averaging multiple samples achieves a lower MSE by approximating the posterior mean, which we use when reporting performance (PSNR values). The number of samples $n$ is set to $n = 2$ when optimizing the linear measurement to optimize performance, as we found empirically that using more samples for the MMSE estimates during optimization does not improve performance during testing. For reporting performance on test data, we used $n = 16$.

### E.4   Optimizing linear measurement

We search for the optimal linear measurement by performing stochastic gradient descent in the space of orthogonal matrices as described in the main text. The linear inverse procedure Algorithm 2 is end-to-end differentiable, and thus optimization of the measurement matrix can be done directly in PyTorch by taking derivatives of reconstruction loss with respect to the matrix parameterization. In all cases, we used the Adam optimizer (Kingma & Ba, 2014) with a learning rate of $10^{-4}$ with exponential decay of 0.90. We used a batch size of 64, and training was run for 16 epochs. The optimization was done on a single node in a GPU cluster, with 4 NVIDIA A100 GPUs. The optimization for a single OLM requires a few hours for the MNIST dataset, and about 24 hours for the CelebA dataset. The computational cost is primarily due to backpropagation through the diffusion model sampling procedure, which involves iterative applications of the U-Net denoiser.

### E.5   Subspace distance

To quantify the difference between two linear measurement subspaces, we used the distance defined on the Grassmann manifold of subspaces. Concretely, given two linear subspaces $F$ and $G$ defined by the column of two measurement matrices $M_1$ and $M_2$, the principal angles $\theta = (\theta_1, \theta_2, ..., \theta_k)$ are defined sequentially as Björck & Golub (1973):

$$\cos \theta_k = \max_{u_k \in F} \max_{v_k \in G} u_k^T v_k, \quad ||u_k||_2 = 1, ||v_k||_2 = 1$$
$$\text{subject to} \quad u_k^T u_j = 0, v_k^T v_j = 0, \ \forall j = 1, 2, ..., k-1. \tag{15}$$

The principle angles can be computed numerically using the singular value decomposition as described in Knyazev & Argentati (2002). The Grassmann distance is defined as the $L_2$ norm on the vector of principal angles $\theta$:

$$d_k(F, G) = (\sum_{i=1}^{k} \theta_i^2)^{1/2}. \tag{16}$$

### E.6 Sequential OLM

The algorithm presented in Section 2.4 learns an OLM independently for each value of $k$. Empirically, we observe that the linear measurement subspace of smaller $k$ is typically not fully contained in that of larger $k$. In this section, we modify the algorithm into a greedy version, where measurement vectors are added one at a time, each conditioned on the previous measurement matrix. This results in a sequentially ordered set of OLMs, analogous to the structure of PCA.

The algorithm works as follows: Given a measurement matrix $M_k \in R^{d \times k}$, we find a set of vectors $U = \{u_1, u_2, ..., u_{d-k}\}$ as a basis of the null space of $M_k$. The new measurement vector $m^*$ is parameterized as $m^* = U\lambda$, where $\lambda$ is a unit vector. The next measurement matrix $M^* \in R^{d \times (k+1)}$ is obtained by concatenating $M_k$ with $m^*$ as $M^* = [M_k \ m^*]$. The objective function in Equation (8) can be written with respect to $\lambda$ as:

$$\mathcal{L}(\lambda) = \frac{1}{N} \sum_{i=1}^{N} ||h(M^*(\lambda)^T x_i; \ M^*(\lambda)) - x_i||_2^2. \tag{17}$$

To obtain a complete set of sequential OLM (denoted as $\text{OLM}_{\text{seq}}$), we start with $k_0 = 1$, and run the greedy algorithm iteratively, increasing the number of measurements $k$ by one at each step.

## F  Supplementary Figures and Tables

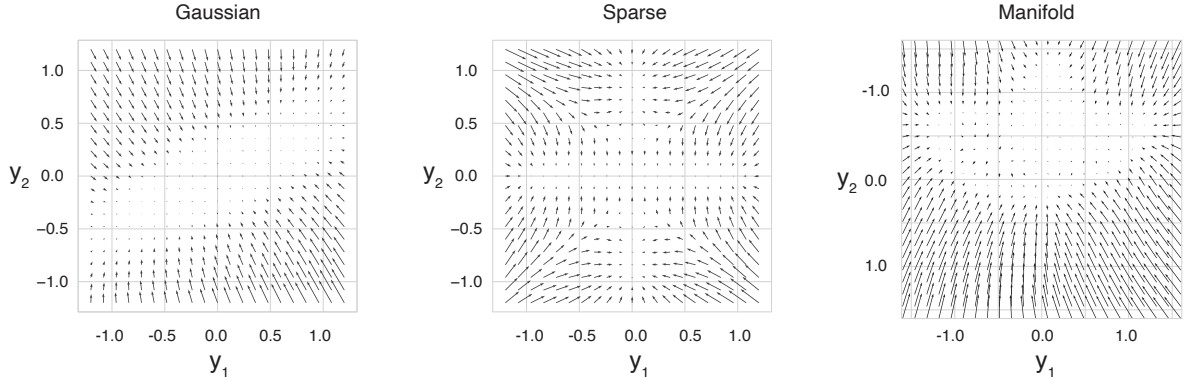

Supplementary Figure 2: Denoiser implicit prior, $\hat{x}(y) - y = \sigma^2 \nabla \log p_\sigma(y)$, depicted as a vector field, for the three different data distributions of Figure 1. Each vector indicates the direction in which the probability of the noisy density increases most rapidly (see Eq. 1).

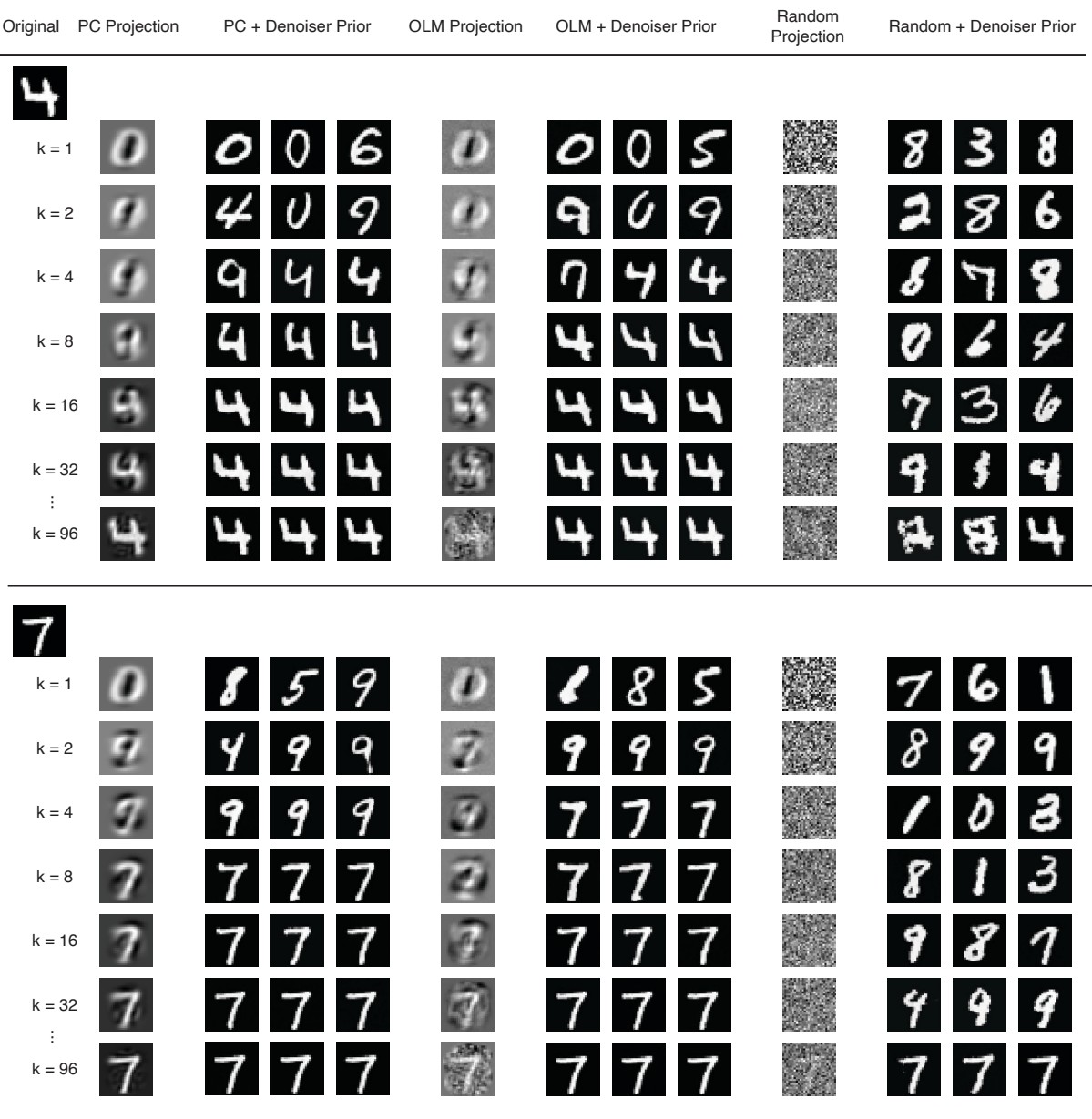

Supplementary Figure 3: Individual samples of the linear inverse estimates from the denoiser prior for two example digit images in the MNIST dataset. The leftmost columns show the PC projection. The next three columns show individual stochastic samples from the denoiser prior, conditioned on the PC measurements. The middle set of columns show the OLM projection, and the samples conditioned on the OLMs. The last set of columns show the results from a random measurement matrix. Rows correspond to increasing number of measurement $k$.

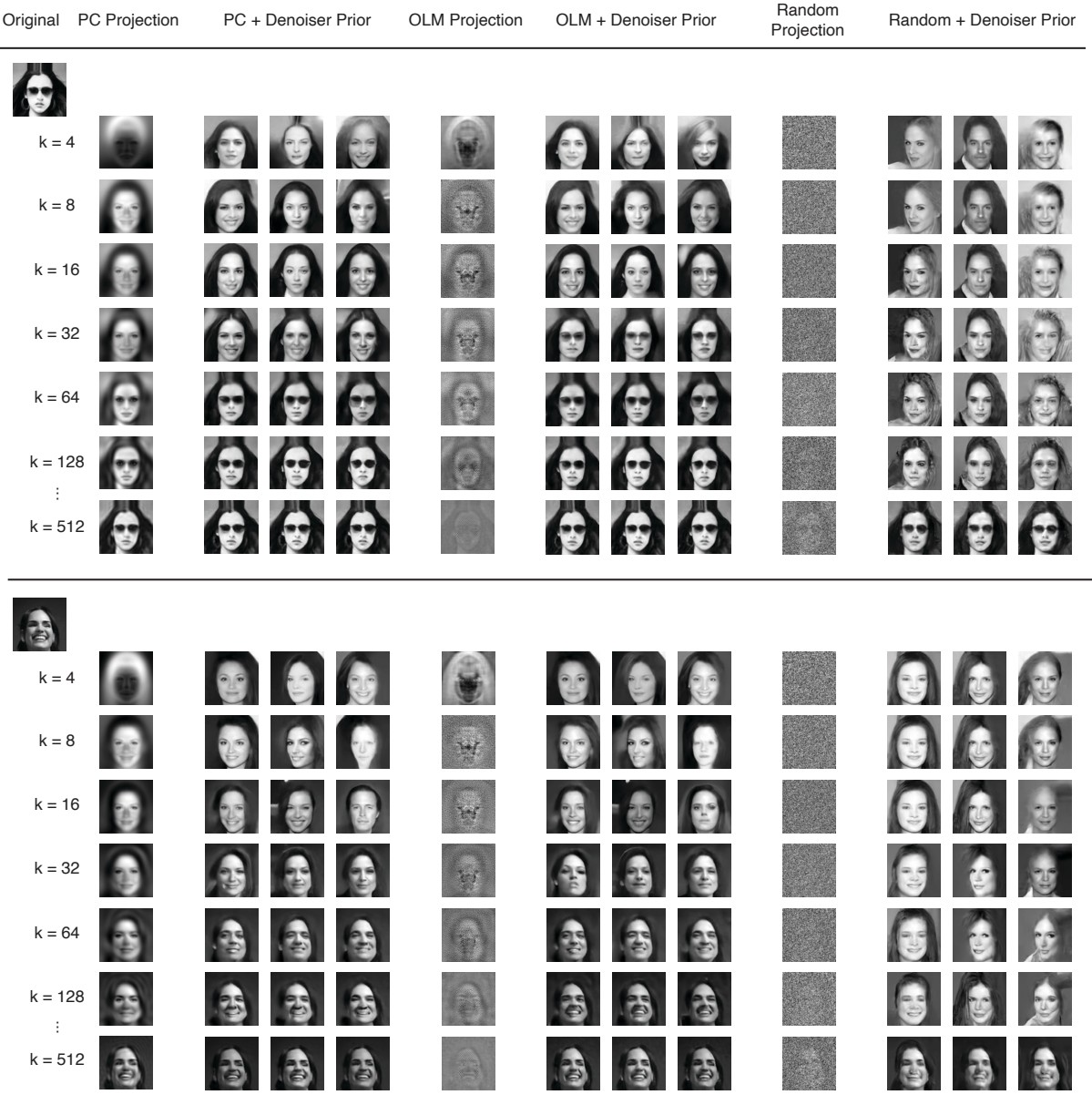

Supplementary Figure 4: Individual samples of the linear inverse estimates from the denoiser prior for two example digit images in the CelebA dataset. The leftmost columns show the PC projection. The next three columns show individual stochastic samples from the denoiser prior, conditioned on the PC measurements. The middle set of columns show the OLM projection, and the samples conditioned on the OLMs. The last set of columns show the results from a random measurement matrix. Rows correspond to increasing number of measurement $k$.

| DATASET | METHODS | Bora et al. (2017) | | Wu et al. (2019b) | | Our Method | |
|---|---|---|---|---|---|---|---|
| | | LASSO (Wavelet) | VAE or GAN Prior | Learned Linear* | Learned *Non-Linear** | PC + Prior | OLM + Prior* |
| MNIST, k = 25 | | 103 | 17.2 | $4.0 \pm 1.4$ | $3.4 \pm 1.2$ | $4.31 \pm 0.13$ | $\mathbf{3.47 \pm 0.11}$ |
| MNIST, k = 50 | | 86 | 8.9 | / | / | $1.59 \pm 0.05$ | $\mathbf{1.20 \pm 0.03}$ |
| MNIST, k = 250 | | 49.1 | 6.2 | / | / | $0.13 \pm 0.01$ | $\mathbf{0.04 \pm 0.002}$ |
| CelebA, k = 50 | | $\sim 235$ | $\sim 129$ | $\sim \mathbf{32.0 \pm 6.7}$ | $\sim 28.9 \pm 6.7$ | $39.31 \pm 0.86$ | $38.29 \pm 0.84$ |
| CelebA, k = 100 | | $\sim 197$ | $\sim 76$ | / | / | $25.51 \pm 0.61$ | $\mathbf{24.77 \pm 0.58}$ |
| CelebA, k = 500 | | $\sim 102$ | $\sim 36$ | / | / | $9.75 \pm 0.27$ | $\mathbf{9.01 \pm 0.26}$ |

Table 1: Compare reconstruction performance to different compressed sensing methods. We compare the per-image MSE (lower is better) of our method with previous methods for compressed sensing using deep neural networks (Bora et al., 2017; Wu et al., 2019b). Methods with symbol "*" use measurement function that are explicitly optimized; The symbol "/ " indicates the results are not available, as Wu et al. (2019b) reported performance only for small $k$; The symbol "$\sim$" indicates the MSE are estimated from images of different size (64 x 64) assuming a constant per-pixel error. In addition, $\pm$ indicates standard error of the mean. Bold numbers highlight the best result for *linear* measurement.

We compare the per-image MSE of our method with previously proposed methods for compressed sensing using deep neural networks (Bora et al., 2017; Wu et al., 2019b). Bora et al. (2017) used both VAE and GAN as image priors to obtain the linear inverse solution. Their methods outperform standard method for compressed sensing using LASSO (sparse prior) in the wavelet domain. They did not attempt to optimize the measurement matrix, and as a result were vastly outperformed by our method.

Wu et al. (2019b) optimized both linear and nonlinear measurements using the restricted isometry property as an objective, but only for very small $k$. We found that on MNIST, our OLM is able to match the performance of even the *nonlinear* measurement function. On CelebA dataset, for $k = 50$, we obtained slightly worse performance for the OLMs compared to the learned linear measurements in Wu et al. (2019b). However, we may have underestimated the error in Wu et al. (2019b), as we estimated MSE based on images of a smaller size (64 x 64) from their results but assumed a constant per-pixel error. Furthermore, our method is most effective for $k$ at around 10% of the total number of pixels. Thus, we expect our advantage to improve further for larger $k$. Lastly, our method has the distinct feature that we can produce individual high-probability samples, in addition to the MMSE solution.

