# OpenReview forum: "Generalized Compressed Sensing for Image Reconstruction with Diffusion Probabilistic Models"
_TMLR — Accepted by TMLR_

### Review · Reviewer_oCSD · 2025-02-28

**Summary Of Contributions:**

This paper considers the problem of learning a linear measurement operator to minimize reconstruction error when using a diffusion prior. In particular, they consider a denoiser prior in which one can then use a conditional sampling algorithm to generate estimates or find an MMSE estimate. Over a training set of images, one optimizes a loss over the space of orthogonal matrices to find an optimal projection such that the posterior mean estimate is as close to the true underlying image as possible. While an ell2-norm is a natural choice to measure discrepancy, the authors also consider different loss functions such as SSIM and perceptual metrics. The authors show that their method, OLM, consistently outperforms PCA and random measurements on several datasets and provide interesting findings on the different types of operators learned using different reconstruction metrics.

**Audience:**

Yes

**Broader Impact Concerns:**

None.

**Claims And Evidence:**

Yes

**Requested Changes:**

I am requesting the following two changes, which I believe would strengthen the work in my view.

*Experimentation on two facets:* I think it would be very interesting to see what the differences are between the current approach and a sequential greedy approach to learn the OLM, leading to a hierarchical set of measurement vectors. Moreover, adding noise to this objective feels quite natural and it would also be interesting to see how such OLMs change with different types of noise (e.g., if one added Gaussian noise vs Poisson).

*Prior literature discussion:* Some prior literature is missing from the discussion, e.g., there are other works that have tried to learn measurement matrices for compressed sensing, but instead use \ell_1 decoders for their reconstruction algorithm, such as

Wu et al, Learning a Compressed Sensing Measurement Matrix via Gradient Unrolling, ICML 2019

There are also other works in the MRI literature that have aimed to learn subsampling strategies by, e.g., using reinforcement learning:

Pineda et al, Active MR k-space Sampling with Reinforcement Learning, MICCAI 2020

It could be good to talk about more recent learning-based methods to learn measurement matrices and discuss the pros and cons.

**Strengths And Weaknesses:**

*Strengths:*
- To the reviewer's knowledge, learning optimal linear measurements for a data-driven diffusion prior is novel. In particular, I am unaware of prior methods focusing on learning measurements for the prior embedded in denoisers.
- The proposed framework provides nice flexibility in determining what "optimal" means by allowing different loss functions to be used, encouraging either low reconstruction error or high perceptual similarity.
- The results provide nice geometric intuition for how the OLM behaves and the authors do a good job of analyzing various properties of the learned set of measurements.

*Weaknesses:*
- Focusing on the mean estimate requires a fair amount of computation, as you would need to run the Langevin-style algorithm multiple times and then take an average to get a single estimate.
- The way the method is proposed does not allow for a type of ordering of the optimal linear measurements. That is, for each fixed k, the set of measurement vectors could be different. This is somewhat of a limitation on interpretability of the resulting measurement vectors.
- Noise could be naturally integrated into the proposed framework (e.g., one could add a fixed amount of Gaussian noise to M^T_k x in the loss function), but it was not considered here.

---

> ### Author Response · Authors · 2025-04-05
> **Response to Reviewer oCSD**
>
> We would like to thank the reviewer for their positive and constructive feedback. Please see our detailed response below.
>
> *Weaknesses*
> > Focusing on the mean estimate requires a fair amount of computation, as you would need to run the Langevin-style algorithm multiple times and then take an average to get a single estimate.
>
> This is a fair point. However, we would like to note that our algorithm performs well even with a single sample; in fact, the single-sample reconstruction generally have better visual quality (see Appendix F). Nonetheless, to conduct a principled comparison with PCA, CS, and other methods using MSE, we approximated the MMSE solution by averaging over reconstructed images.
>
> In addition, in our algorithm implementation, we treat the number of samples as the batch dimension in the U-Net denoiser, and thus can obtain many samples in one Langevin-style run. As a result, in practice, generating more samples only marginally increases the computation time.
>
> > The way the method is proposed does not allow for a type of ordering of the optimal linear measurements. That is, for each fixed k, the set of measurement vectors could be different. This is somewhat of a limitation on interpretability of the resulting measurement vectors.
>
> Thanks for the very interesting suggestion. Our algorithm can indeed be modified to a sequential greedy version. We had done some preliminary tests of this before submitting the paper, but had decided that it would complicate the story. We have now incorporated these additional experimental results into Section 3.4 and Figure 4.
>
> We find that adding the sequential constraint results in a marginal decrease in the performance of the OLMs, although they still outperform PCs. We also note that although the greedy measurement functions are different than the original OLMs, they are more similar to the OLMs than the PCs.
>
> > Noise could be naturally integrated into the proposed framework (e.g., one could add a fixed amount of Gaussian noise to M^T_k x in the loss function), but it was not considered here.
>
> We agree that inclusion of measurement noise is an important extension, and we intend to pursue it further in future work. If noise is added to the linear measurement, the reconstruction algorithm should be modified to handle the noise correctly, a direction we are actively working on. As such, fully addressing the broader issue of noisy measurements is beyond the scope of our current paper.
>
> Within our current framework, we can investigate noise robustness of the OLMs compared to the PCs, and whether including measurement noise during optimization can increase noise robustness (even when this is done with a reconstruction algorithm that is not modified to handle noise). We have added these additional experimental results to Section 3.4 and Figure 4. We show we can train an OLM that is more robust to noise, at the cost of a small decrease in reconstruction performance in low- or no-noise regimes.
>
> *Requested Changes*
>
> Experimentation on two facets: Yes, we have included additional experiments as outlined in our response above.
>
> Prior literature discussion: Thanks for pointing us to this literature. We have included them in a revised Related Work section, and expanded our discussion.

---

### Review · Reviewer_o6TM · 2025-03-08

**Summary Of Contributions:**

The paper introduces a novel method for optimizing linear measurements used in compressed sensing image reconstruction. This is achieved by leveraging recent developments on solving inverse problems with diffusion prior and optimizing the measurement matrices for a lower reconstruction loss. Experimental results on both synthetic and real image data demonstrate that the proposed method (OLM) outperforms both principal components (PC) and random measurements when using diffusion priors for reconstruction.  Additionally, the paper provides insightful visualizations and statistical analyses that reveal distinct characteristics between PCs and OLMs.

**Audience:**

Yes

**Broader Impact Concerns:**

None.

**Claims And Evidence:**

Yes

**Requested Changes:**

The manuscript looks great in its current version, but I believe addressing the questions above in more detail could further enhance its clarity and impact.

**Strengths And Weaknesses:**

#### Strengths

- The approach of optimizing linear measurements for compressed sensing using diffusion-based inverse problem solvers is both innovative and effective.
- The experimental results show clear advantages of OLM against both principal components and random projections, which shows the data compressing ability of data-driven prior.
- Analyses on the characteristics of measurements and optimizing image perceptual loss are interesting and may interest a boarder audience.
- The paper is overall well-written and easy to follow.

#### Questions

- How does the optimization of measuring matrices scale with respect to the data dimension and the number of diffusion sampling steps?
- How do OLMs optimized by perceptual loss differ from those optimized by MSE? Specifically, do measurements optimized for SSIM also reduce reconstruction error in terms of PSNR?
- I notice some missing recent works on solving inverse problems with plug-and-play diffusion priors, such as [1,2,3], which seem to claim better performance on image reconstructions. Would the OLMs be different using the same diffusion prior but a different plug-and-play inverse problem solver? Would it be a meaningful direction to explore OLMs with more advanced plug-and-play diffusion samplers?
- The MMSE estimate is calculated "as the average over multiple conditional samples." How does the number of conditional samples used affect the final performance of OLMs?

[1] Wang et al. "Zero-Shot Image Restoration Using Denoising Diffusion Null-Space Model." ICLR 2023.

[2] Zhang et al. "Improving Diffusion Inverse Problem Solving with Decoupled Noise Annealing." arXiv:2407.01521.

[3] Li et al. "Decoupled Data Consistency with Diffusion Purification for Image Restoration." arXiv:2403.06054.

---

> ### Author Response · Authors · 2025-04-05
> **Response to Reviewer o6TM**
>
> We would like to thank the reviewer for their positive and constructive feedback. Please see our detailed response below.
>
> *Questions*
> > How does the optimization of measuring matrices scale with respect to the data dimension and the number of diffusion sampling steps?
>
> Our denoiser is implemented with a U-Net, with cost O(N), or perhaps O(N logN), since we use a U-Net architecture whose depth should generally be increased as the log of the image size N.  Empirically, we found that larger image size N or a larger number of measurements K only slightly increase the number of denoising iterations required by our reconstruction algorithm
> Note that for this model, the number of steps in the diffusion sampling procedure is automatically determined by the fixed threshold parameter sigma_end, regardless of image size.
>
> Here are empirical optimization time (for the same number of epochs) on the MNIST dataset, as a function of K:
> | Number of Measurements (K)     |   32   |   64   |   96   |  128   |  256   |  512   |
> |-------------------------------|--------|--------|--------|--------|--------|--------|
> | Optimization Time (hh:mm)     | 02:24  | 02:22  | 02:31  | 03:25  | 02:46  | 03:20  |
>
> As noted above, we are not planning to revise the paper with a discussion of how computational cost scales, but the discussion in this response will be available through this open review process. If the reviewers feel it is important to include it in the paper, we can do so.
>
>
> > How do OLMs optimized by perceptual loss differ from those optimized by MSE? Specifically, do measurements optimized for SSIM also reduce reconstruction error in terms of PSNR?
>
> We have revised the text to include the following information in Results section 3.5:
>
> We have conducted the same analysis as in Figure 4D, and found that the subspace distance between OLMs optimized toward different objectives is comparable to their distance from the PCs (i.e., they are all distinct from each other).
>
> Furthermore, we note that the OLMs optimized for SSIM result in worse PSNR compared to the PC measurement with denoiser prior reconstruction, suggesting that there is a trade-off between optimizing for perceptual quality versus MSE reconstruction, at least with our methods.
>
> > I notice some missing recent works on solving inverse problems with plug-and-play diffusion priors, such as [1,2,3], which seem to claim better performance on image reconstructions. Would the OLMs be different using the same diffusion prior but a different plug-and-play inverse problem solver? Would it be a meaningful direction to explore OLMs with more advanced plug-and-play diffusion samplers?
>
> We have included the papers mentioned above in the Related Work section. In general, our experience suggests that the performance of OLMs is primarily determined by the quality of the diffusion image prior, rather than the specific details of the reconstruction and sampling algorithm. For example, [1] uses an algorithm conceptually similar to ours, but uses DDPM instead of a Langevin-like sampling procedure. We are interested in exploring this further in future work.
>
> > The MMSE estimate is calculated "as the average over multiple conditional samples." How does the number of conditional samples used affect the final performance of OLMs?
>
> We have added additional results to Appendix Section D to show how different parameters of the algorithm, including the number of conditional samples, affect the reconstruction performance. Note that we used the same conditional averaging procedure with n = 16 for all comparisons, including those with PCs and random measurements.
>
> Basically, using a few conditional samples to estimate the posterior mean significantly improves performance, but this improvement quickly saturates as the number of samples increases.

---

### Review · Reviewer_wxhP · 2025-03-12

**Summary Of Contributions:**

This paper presents an enhanced linear compressed sensing method that leverages a prior model based on the diffusion probabilistic model. By incorporating the constraint sampling method from previous work, the posterior mean under a given linear measurement is more accurately approximated. Building on this improved approximation, the paper formulates an optimization problem to determine the optimal linear measurement. Experimental results on a 2D synthetic dataset, MNIST, and CelebA demonstrate that the proposed method consistently outperforms conventional approaches such as PCA and ICA. This advancement has the potential to enhance practical applications relying on compressed sensing.

**Audience:**

Yes

**Broader Impact Concerns:**

No concern.

**Claims And Evidence:**

Yes

**Requested Changes:**

1. In section 2.3, the definition of the MMSE proposed in (4) is not clear. It was not until I read section E.3 that I realized how the MMSE is estimated. I suggest moving section E.3 to section 2.3 or at least referring to it in section 2.3. It will make the writing more clear.
2. Above equation (7), is there a typo for $Q \in \mathbb{R}^{d \times k}$? From (7), Q seems to be a square matrix.
3. Could the author discuss more about the statement on Page 7 under Figure 3: "This indicates that the union of subspace priors used in compressed sensing literature does not accurately describe the properties of natural images."
4. I have a question related to the optimizing time. Why does it cost a few hours for MNIST and 24 hours for the CelebA dataset to optimize the matrix with only $dk$ variables? Also, with A100 of 80 GB, why could you only use 2 samples to estimate the posterior mean? Where does the cost and computation come from?

**Strengths And Weaknesses:**

## Strength
1. the experiment of this paper, demonstrates that given a non-linear prior, a better approximation of the prior (compared with an approximation like Gaussian) could significantly improve the compressed sensing performance, and even the linear measurement could be benefited. It will inspire future work for compressed sensing areas.
2. experiments in this paper are solid, and clearly demonstrate the proposed compressed sensing method outperforms PCA and ICA.
3. The writing in this paper is well-organized in general, I could follow the entire paper even without much background in compressed sensing.

## Weakness
1. The proposed method is not efficient compared with PCA and ICA, both for 1. optimizing the measurement matrix and 2. using the diffusion model to reconstruct from the measurement, as mentioned by the author in the paper. Also, the cost of training a diffusion model is not negligible.
2. The paper misses the ablation study of the posterior estimation, i.e., the number of samples to estimate the posterior mean, sampling steps for the diffusion model, sampling method, and hyperparameters you proposed in Algorithm 2.
3. I am a little concerned about extending this method to real-world applications. As I mentioned, there is a need for a training diffusion model for this proposed method, which needs enough training data and time. However, for real-world scenarios like cameras, is there enough training data to train the diffusion model? Otherwise, given out-of-domain data from the camera, what's the performance of the proposed method? Therefore, I highly suggest using a large-scale diffusion model, like stable diffusion [1], to test the zero-shot performance of your method on the out-of-domain dataset. This experiment will significantly improve the practical impact of the proposed method.

[1] Rombach, Robin, Andreas Blattmann, Dominik Lorenz, Patrick Esser, and Björn Ommer. "High-resolution image synthesis with latent diffusion models." In Proceedings of the IEEE/CVF conference on computer vision and pattern recognition, pp. 10684-10695. 2022.

---

> ### Author Response · Authors · 2025-04-05
> **Response to Reviewer wxhP**
>
> We would like to thank the reviewer for their positive and constructive feedback. Please see our detailed response below.
>
> *Weakness*
> 1.
> Our method is indeed computationally more expensive than PCA, both in terms of finding the optimal linear measurements and in terms of image reconstruction (when projection is used for PCA reconstruction). For many practical applications, however, the diffusion model, and corresponding OLMs need only be computed once. For reconstruction costs, note that although the PCA projection solution is fast, the quality (figure 3, blue line) is substantially worse than our method.
>
> ICA (and compressed sensing) on the other hand, also require an iterative reconstruction algorithm. The complexity of this algorithm is of order O(KN), with K being the number of measurements and N the total number of pixels.  In contrast, the computational cost of applying our denoiser is O(N), or perhaps O(N logN), since we use a U-Net architecture whose depth should generally be increased as the log of the image size.  Empirically, we found that a larger image size N or a larger number of measurements K only slightly increase the number of denoising iterations required by our reconstruction algorithm (see our reply to Reviewer o6TM). Thus, we believe our diffusion reconstruction algorithm will be more efficient than ICA reconstruction.
>
> We are not currently planning to include a discussion of how computational cost scales in the revision, as the discussion here is available through the open review process. If the reviewers feel it is essential to include this discussion in the paper, we can do so.
>
> 2.
> This is a great suggestion. We have added additional results to Appendix Section D describing the impact of these parameters on reconstruction performance.
>
> 3.
> We agree that out-of-domain generalization is an important and interesting question for diffusion models. However, in the context of compressed sensing and optimal measurement, the primary focus is typically on within-domain performance. For instance, in applications like medical imaging or computational microscopy, the goal is to maximize measurement efficiency with the knowledge that the signals originate from a constrained domain, instead of generalizing the measurements to arbitrary signals. We feel that careful consideration of this issue is best reserved for future work.
>
> *Requested Changes*
> 1.
> Thanks for the suggestion. We have revised the text accordingly and now also refer readers to E.3 for more detail.
>
> 2.
> Q is of shape d×k as it is the measurement matrix. We mistakenly called Q an orthogonal matrix. We have revised the description: “Concretely, we parameterize the set of matrices Q ∈ Rd×k with orthonormal columns using the Householder product.”
>
> 3.
> Thank you for the question. We have revised the text surrounding Figure 3 to better clarify this point.
>
> The main idea here is that 1) If the natural image prior truly follows a union of subspace (sparse) prior, we expect the diffusion model should be expressive enough to capture it, and 2) the compressed sensing literature has shown that random projections are very effective measurements for signals with a union of subspaces prior. Under these conditions, applying our reconstruction to random projection measurements should be near-optimal.  However, we observe that reconstruction from random projection measurements is significantly worse than from OLMs, and is even outperformed by principal components (PCs) (see Fig 3). This suggests that images do not lie on a union of subspaces. We also note that while this point has been made before in the literature (e.g., Weiss et al., 2007), the fact that the diffusion model is an extremely good model of images helps us reach this conclusion definitively, while providing a better measurement solution.
>
> 4.
> The computational cost is primarily due to the diffusion model sampling procedure, which involves iterative applications of the denoiser.
>
> Additionally, we realized that our original description of the sample size was misleading. The memory requirement during training is determined by (batch size * sample size). Empirically, we found that using more than two samples for the MMSE during optimization does not improve performance during testing (where we use n=16). Therefore, we set the sample size to 2 to enable the use of a larger batch size.
>
> We have revised Appendix E.3 and E.4 to clarify these points.

---

### Author Response · Authors · 2025-03-26
**We are currently working on revising the paper**

We would like to thank the reviewers for their positive assessment and constructive feedback.

We are unclear (and perhaps the editor can clarify) whether we are expected to provide a revised manuscript as part of this discussion phase, or whether submission and evaluation of a revision would follow, should a revision be invited.

In any case, we will provide a detailed response to the points raised by each reviewer in the next few days, including our plans for revision. Some of the additional experiments in our revision will take more time to complete, but we are actively working on them now.

---

> ### Comment · Action_Editor_xgMB · 2025-03-31
> **Clarification on Revised Manuscript Submission**
>
> Dear authors,
>
> Thank you for your inquiry. You are welcome to submit a revised manuscript during the discussion period.
>
> Best regards,
>
> AE

---

### Author Response · Authors · 2025-04-05
**Overall Response & Revision Summary**

We thank the reviewers for their constructive feedback. In response to the comments, we have introduced improvements throughout the main text, and we have incorporated the following new results:

1) Section 3.4, Figure 4G, and Appendix E.6: We show the performance of a greedy version of the algorithm, in which the OLMs are sequentially orthogonalized (analogous to PCA).  Performance is only slightly worse than the nonsequential (joint) algorithm.

2) Section 3.4, Figure 4H: We provide a simple analysis and demonstration that reconstruction from OLM measurements is reasonably robust to noise, and can be made more so by training on noisy measurements.

3) Appendix D: We provide additional experiments showing the effects of hyperparameter choices (step size, injected noise, stopping criterion, and number of samples) on OLM reconstruction performance.

We have uploaded a revised version of the manuscript, and provided responses to the specific comments of each reviewer below.

---

### Decision · Action_Editor_xgMB · 2025-04-25

**Recommendation:** Accept as is

**Comment:**

After a thorough evaluation of the paper, I recommend **acceptance**.

### Summary and Strengths

This paper presents a compelling framework for learning optimal linear measurements tailored to the priors embedded in diffusion probabilistic models. The central idea, jointly optimizing a measurement matrix to minimize reconstruction error via denoiser-guided diffusion, is well-motivated and executed with rigor. Key strengths include:

- **Empirical Validation**: Extensive experiments on MNIST and CelebA show consistent performance gains over PCA, ICA, and random projections under both MSE and SSIM-based loss functions.
- **Thorough Analyses**: The authors investigate the impact of sampling parameters, number of measurements, noise robustness, and sequential optimization, reinforcing the robustness of the approach.
- **Reviewer Consensus**: All three reviewers recommend acceptance and find the claims supported by convincing evidence. Their comments raised several important points that prompted meaningful author revisions, including improvements in clarity (e.g., MMSE definition and matrix orthogonality), additional experiments using a sequential (greedy) optimization algorithm and robustness to noisy measurements, and better positioning with respect to prior work. The authors engaged constructively with all feedback, and the discussion phase helped significantly enhance the quality and clarity of the final manuscript. Several constructive points were raised and addressed, including computational cost, generalizability, and missing literature.

### Points of Discussion

Some concerns were raised around generalizability and practicality:

- **Computational Cost**: The optimization process is more intensive than PCA or ICA. However, the authors justify this by highlighting the benefits in reconstruction quality and the once-off nature of the optimization.
- **Out-of-Domain Generalization**: Reviewer wxhP suggested testing with large-scale pretrained diffusion models like Stable Diffusion. While a valid future direction, the authors appropriately focus on within-domain scenarios, aligning with typical compressed sensing use cases.

The authors also made decent efforts to improve the paper during the discussion phase, including:

- Adding results on greedy (sequential) optimization and noise robustness.
- Expanding the related work section to include more recent literature.
- Clarifying definitions and revising unclear sections (e.g., MMSE estimation and matrix orthogonality).

**Audience:**

The paper addresses a core problem in machine learning and signal processing (linear inverse problems) using modern tools from generative modeling. The proposed method will likely interest both researchers developing image reconstruction algorithms and those studying the use of learned priors in inverse problems. The methodology and results are accessible to the TMLR audience and relevant to the broader ML and computer vision communities.

**Claims And Evidence:**

The submission introduces a novel framework for optimizing linear measurements for image reconstruction under diffusion probabilistic models. The authors support their claims with strong empirical evidence across synthetic and real datasets (MNIST and CelebA). The reconstruction results consistently outperform standard methods like PCA and ICA, and the experimental design clearly supports the paper’s central thesis. Additional robustness analyses and sensitivity studies further strengthen the evidence.

---

> ### Author Response · Authors · 2025-05-09
> **Thank everyone for the great feedback**
>
> We would like to once again thank the reviewers and the editor for their constructive feedback. It significantly improved the paper, and also points to interesting directions for future research.
>
> We have uploaded the camera-ready version of the paper, incorporating some minor revisions throughout.